# New Gd^3+^ and Mn^2+^-Co-Doped Scheelite-Type Ceramics—Their Structural, Optical and Magnetic Properties

**DOI:** 10.3390/ijms232415740

**Published:** 2022-12-12

**Authors:** Hubert Fuks, Paweł Kochmański, Elżbieta Tomaszewicz

**Affiliations:** 1Department of Technical Physics, Faculty of Mechanical Engineering and Mechatronics, West Pomeranian University of Technology in Szczecin, Al. Piastów 19, 70-310 Szczecin, Poland; 2Faculty of Mechanical Engineering and Mechatronics, West Pomeranian University of Technology in Szczecin, Al. Piastów 19, 70-310 Szczecin, Poland; 3Department of Inorganic and Analytical Chemistry, Faculty of Chemical Technology and Engineering, West Pomeranian University of Technology in Szczecin, Al. Piastów 42, 71-065 Szczecin, Poland

**Keywords:** scheelites, calcium molybdate, Mn^2+^ and Gd^3+^ ions, UV-vis spectroscopy, EPR spectroscopy

## Abstract

New Gd^3+^- and Mn^2+^-co-doped calcium molybdato-tungstates with the chemical formula of Ca_1−3*x*−*y*_Mn*_y_*▯*_x_*Gd_2*x*_(MoO_4_)_1−3*x*_(WO_4_)_3*x*_ (labeled later as CaMnGdMoWO), where ▯ denotes vacant sites in the crystal lattice, 0 < *x* ≤ 0.2500 and *y* = 0.0200 as well as 0 < *y* ≤ 0.0667 and *x* = 0.1667 were successfully synthesized by high-temperature solid-state reaction method and combustion route. Obtained ceramic materials crystallize in scheelite-type structure with space group *I*4_1_/*a*. Morphological features and grain sizes of powders under study were investigated by SEM technique. Spectroscopic studies within the UV-vis spectral range were carried out to estimate the direct band gap (*E*_g_) and Urbach energy (*E*_U_) of obtained powders. EPR studies confirmed the existence of two types of magnetic objects, i.e., Mn^2+^ and Gd^3+^ ions, and significant antiferromagnetic (AFM) interactions among them.

## 1. Introduction

Divalent metal molybdates and tungstates with the chemical formula of AMo(W)O_4_ form a wide and important family of inorganic materials that have high potential applications in many fields such as electronics and optoelectronics [1,2,3,4,5,6]. They are used as phosphors, lasers matrices, and scintillator detectors [1,2,3,4,5,6]. These compounds with relatively large cations (ionic radius of A^2+^ > 0.99 Å, e.g., Ca^2+^, Sr^2+^, Ba^2+^, and Pb^2+^) exist in scheelite-type structure (tetragonal symmetry, space group *I*4_1_/*a*; Z = 4) where molybdenum or tungsten ions adopt a tetrahedral coordination, while divalent metal ions represent an eight-coordinated position (Figure 1) [7,8]. Due to high thermal stability as well as a relatively low cost of fabrication scheelite-type ceramic molybdates and tungstates are excellent hosts for *d*- and *f*-electron ions.

Rare earth (RE^3+^)-doped molybdates and tungstates have been used extensively mainly in optoelectronics as efficient phosphors, lasers, and scintillators [6,9,10]. The development of wireless communication technologies observed for many years (5G telecommunication and IoT technology) has also resulted in growing interest in advanced ceramic dielectric materials which are usually complex oxide systems. Doped scheelite-type molybdates and tungstates turned out to be excellent microwave dielectrics that could be used in the production of resonators, filters, and antennas [4,11,12].

Gadolinium, one of the rare earth metal families, plays an important role in chemistry and biomedicine. It can be applied simultaneously to magnetic resonance imaging (MRI), X-ray computed tomography (CT), and neutron capture therapy for cancers. Gadolinium ion (Gd^3+^) has tremendous importance, because of its high spin magnetic moment due to seven unpaired electrons (^8^S_7/2_). Therefore, materials doped with Gd^3+^ ions can be applied in magnetic resonance imaging [13]. A very high X-ray absorption coefficient of Gd^3+^ is responsible for the application of these materials in X-ray computed tomography [13,14,15,16,17,18].

Our research group has been conducting studies on new doped micro- and nanomaterials of scheelite-type structure for many years [19,20,21]. We demonstrated that the substitution of such divalent ions as Ca^2+^ or Pb^2+^ in Cd(Pb)Mo(W)O_4_ by trivalent RE^3+^ ones leads to the formation of vacant sites in these matrices (denoted as ▯) what is very well observed by many spectroscopic techniques. Above mentioned RE^3+^-doped scheelite materials revealed very strong luminescence both in visible and NIR spectral ranges that could be suitable for ultra-short pulses lasers and efficient phosphors [22,23]. Dielectric studies of Gd^3+^-doped scheelite-type molybdates and molybdato-tungstates provided much important information on their properties and potential applications [22,23]. These materials are insulators with a residual electrical *n*-type conduction not extending 2·10^−9^ S/m and a direct (or indirect) band gap energy higher than 3 eV [24,25,26,27]. The chemical compatibilities of Gd^3+^-doped micro- and nanomaterials with metallic aluminium and silver as intrinsic electrodes make them suitable for LTCC (Low-Temperature Co-fired Ceramic) technology [26]. Some of them, e.g., Pb_1−3*x*_▯*_x_*Gd_2*x*_(MoO_4_)_1−3*x*_(WO_4_)_3*x*_ where 0 < *x* ≤ 0.1774 exhibit electrical relaxation processes [26,27]. These properties of scheelite-type materials lead to the conclusion that they are very good candidates for the fabrication of lossless capacitors.

Doping is a common procedure to optimize materials for specific industrial applications. The combination of several dopants (called also co-doping) is a very promising way of enhancing many properties of materials such as photoluminescence, and magnetic and dielectric properties.

For this reason, in this work, we have obtained new Gd^3+^- and Mn^2+^-co-doped scheelite-type calcium molybdato-tungstates using two different synthesis methods. Their structure, morphological features, and thermal stability were systematically investigated and analyzed. The optical and magnetic properties of these materials were compared as a function of both dopant’s concentration.

## 2. Results and Discussion

### 2.1. X-ray Diffraction Studies of CaMnGdMoWO Materials

Powder XRD patterns of CaMoO_4_ and CaMnGdMoWO samples with different Gd^3+^ ions content, i.e., when 0 < *x* ≤ 0.2500 and *y* = 0.0200 as well as with different Mn^2+^ concentration, i.e., when 0 < *y* ≤ 0.0667 and *x* = 0.1667 obtained by solid-state reaction method and combustion route are shown in Figure 2a–f. The XRD studies revealed that these patterns consisted of diffraction peaks that can be attributed to scheelite-type framework only. No other phases, i.e., initial reactants, metal oxides, and other gadolinium tungstates or molybdates were observed in CaMnGdMoWO ceramics after their synthesis. The observed diffraction lines attributed to scheelite-type structure shifted slightly towards a lower *2Θ* angle with increasing Gd^3+^ content (Figure 2b,f) or a higher *2Θ* angle with increasing Mn^2+^ concentration (Figure 2d). In order to show the changes in the position of diffraction peaks, the enlarged portion of XRD patterns from 28 to 29.5° *2Θ* with the (112) and (103) diffraction lines are presented in Figure 2b,d,f. All registered diffraction peaks were successfully indexed to pure tetragonal scheelite-type structure with space group *I*4_1_/*a* (No. 88, CaMoO_4_—JCPDs No. 04-013-6763). Table A1 shows the calculated lattice parameters of CaMoO_4_ and doped samples. To observe the changes of unit cell parameters clearly, cell constants (*a* and *c*), lattice parameter ratio *c*/*a* as well as the volume of unit cell (*V*) of CaMnGdMoWO materials obtained by solid-state reaction method and combustion route as a function of Gd^3+^ concentration are shown in Figure 3a,b, and Figure 4a,b, respectively. The observed changes of *a* and *c* lattice parameters are not identical (Figure 3a and Figure 4a). Initially, when Gd^3+^ content is relatively low, the *a* parameter increases up to 5.23958(6) Å (*x* = 0.1430, synthesis by solid state reaction) or 5.2374740(8) Å (*x* = 0.1667, combustion) and then, its value decreases and then increases again with further increasing content of gadolinium ions. The *c* parameter shows a different relationship, i.e., for low Gd^3+^ ions concentrations its value decreases to 11.4173(7) Å (*x* = 0.0839, solid-state reaction method) or 11.4188(9) Å (*x* = 0.0455, combustion). With a further increase of Gd^3+^-doping, the value of *c* parameter increased and then decreased (high-temperature sintering) or only increased (combustion route). Generally, the unit cell volume tends to increase with increasing Gd^3+^ ion concentration, i.e., practically within the whole homogeneity range of CaMnGdMoWO solid solution an expansion of crystal lattice is observed (Figure 3b and Figure 4b). This is not typical behavior because bigger Ca^2+^ ions in the CaMoO_4_ matrix were simultaneously substituted by much smaller Gd^3+^ and Mn^2+^ ones (ionic radius of Ca^2+^ (1.12 Å) > ionic radius of Gd^3+^ (1.053 Å) > ionic radius of Mn^2+^ (0.96 Å) for CN = 8) [28]. In scheelite-type divalent metal molybdates and tungstates with the chemical formula of AMo(W)O_4_ (A = Ca, Sr, Ba, and Pb) Mo^6+^ and W^6+^ ions are tetrahedrally coordinated by oxygen ones and their ionic radii are very similar, i.e., they are 0.41 and 0.42 Å, respectively [28]. Therefore, a substitution of Mo^6+^ ions with W^6+^ ones in scheelite-type structure does not cause significant changes in both lattice parameters of materials. The crystal lattice expansion observed for CaMnGdMoWO materials when Gd^3+^ ions concentration was increasing is probably caused by a strong defect of their structure, i.e., when three crystallographic positions of calcium ions are occupied by two Gd^3+^ ones only and an appearance of cationic vacants takes place. An analogous phenomenon in other doped materials, e.g., RE^3+^-doped CdMoO_4_ (RE = Eu, Dy) [23,24], Eu^3+^ and Mn^2+^-co-doped calcium molybdato-tungstates [29] have already observed. The lattice parameter ratio *c*/*a* was also calculated. This parameter evidently changes nonlinearly with increasing of Gd^3+^ ions concentration (Figure 3b and Figure 4b, Table A1). This means that in the whole homogeneity range of solid solution under study, the deformation of tetragonal scheelite cell of CaMnGdMoWO materials is observed.

Moreover, we have found that the lattice parameters (both *a* and *c*) as well as the volume of a tetragonal unit cell of CaMnGdMoWO ceramics decrease with increasing Mn^2+^ concentration when Gd^3+^ ions content in these samples is constant (*x* = 0.1667, Table A1). Contrary to the materials with a constant of Mn^2+^ concentration (*y* = 0.0200) and variable Gd^3+^ ions content (0 < *x* ≤ 0.2500), calculated lattice parameters and volume of unit cell obey the Vegard’s law, i.e., they are nearly linear functions of *y* concentration parameter.

The density of Gd^3+^ and Mn^2+^-co-doped calcium molybdato-tungstates increases linearly with the increase of both dopant’s content (Table A1).

### 2.2. Thermal Stability and Morphology of CaMnGdMoWO Solid Solution

Our earlier studies showed that both calcium scheelites, i.e., CaMoO_4_ (*powellite*) and CaWO_4_ (*scheelite*) melt congruently at 1480 and 1590 °C, respectively [30]. The melting points of Gd^3+^ and Mn^2+^-co-doped calcium molybdato-tungstates are lower than the melting point of calcium molybdate and they gradually decreased with both increasing of Gd^3+^ and Mn^2+^ ions contents. These temperatures were determined as: 1430 °C (*x* = 0.0050 and *y* = 0.0200), 1290 °C (*x* = 0.1667 and *y* = 0.0200), 1260 °C (*x* = 0.2500 and *y* = 0.0200), and 1260 °C (*x* = 0.1667 and *y* = 0.0667).

Figure 5 shows SEM images of CaMnGdMoWO ceramic materials (*x* = 0.1667 and *y* = 0.0200) obtained by two different routes, i.e., high-temperature sintering (Figure 5a) and combustion method (Figure 5b). The sample of the solid solution obtained by solid state reaction method exhibits well-defined and sharp grain boundaries. It suggests that this material obtained by multi-hour annealing at high temperatures is well-crystallized. This sample contains irregular spherical-like particles that a grain size ranging from ~5 to ~20 μm. Occasionally, big grains and multi-grain agglomerates reaching ~50 µm appear. The CaMnGdMoWO ceramic material of the same composition obtained by the combustion method shows a different morphology. It is composed of unform spherical particles with the average size ranging from ~500 nm to a few micrometers (Figure 5b). The smaller grain size of CaMnGdMoWO samples (submicro) obtained by the combustion method was also confirmed by XRD results. Figure 6 shows powder XRD patterns of samples with the same composition, i.e., when *x* = 0.2500 and *y* = 0.0020 but obtained by two different methods.

The diffraction lines observed for all materials obtained by the combustion route are clearly wider and less intense than the ones registered for samples obtained by long-term high-temperature sintering. EDS elemental analysis was employed to confirm both the purity and chemical composition samples under study. EDS spectra (Figure 5) revealed that only elements which were present in CaMnGdMoWO materials were Ca, Mn, Gd, Mo, W, and O. No peaks of any contaminations were detected suggesting the high purity of all obtained samples. All identified elements were evenly distributed throughout the surface of the samples under study revealing a uniform chemical composition of each material.

### 2.3. Optical Properties of CaMnGdMoWO Ceramic Materials

Measurement of diffuse reflectance with a UV-visible spectrophotometer is the standard technique in determination of absorption properties of solid materials. In the case of doped materials, the properties that can potentially be estimated from diffuse reflectance are band gap energy, absorption coefficient, and refractive index. These optical properties are very important parameters characterizing materials for optoelectronic applications, e.g., photocatalysts, efficient lasers, and scintillators as well as in solar cells. Band gap energy is also the primary factor determining electrical conductivity of solids.

Generally, there are two types of optical transition that can occur at the fundamental edge of crystalline materials: direct and indirect transitions [31,32,33,34]. Both involve the interaction of an electromagnetic wave with an electron in a valence band, which is raised across the fundamental gap to a conduction band. However, indirect transitions also involve simultaneous interaction with lattice vibrations. Thus, the wave vector of the electron can change in the optical transition, the momentum change being taken or given up by phonons. According to the literature information, most of divalent metal molybdates and tungstates with tetragonal scheelite-type structure (e.g., CaMoO_4_ and CaWO_4_) exhibit an optical absorption spectrum governed by a direct transition, i.e., an electron located in a maximum-energy state in a valence band achieves a minimum-energy state in a conduction band under the same point in the Brillouin zone [31,32,33,34].

Optical energy gap (E_g_) of CaMoO_4_ and CaMnGdMoWO materials was determined by the method proposed by Kubelka and Munk [35] and applied by us in our earlier studies [24,30,36]. This methodology is based on a transformation of diffuse reflectance spectra into absorption ones to estimate E_g_ values. The UV-vis absorption spectra of CaMoO_4_ and CaMnGdMoWO ceramics obtained by solid-state reaction as well as combustion methods are shown in Figure 7a and Figure 8a, respectively. The intense and broad absorption bands within the spectral range of 200–375 nm can be ascribed to O^2–^ → W^6+^, O^2–^ → Mo^6+^ as well as O^2–^ → Mn^2+^ charge transfer bands [9,11,22,23,24,29,30]. The trivalent gadolinium ion, Gd^3+^ has half-filled 4f orbitals (4f^7^ configuration, ground state ^8^S_7/2_); therefore, it is expected that the intensity of absorption bands due to 4f-4f transitions would be very weak. All of these bands should be observed in the very high-energy part of the spectrum, i.e., within the UV region (195–310 nm) [20,26]. So, they are covered by CT bands in the UV-vis spectra of CaMnGdMoWO samples.

When a structure of a band gap is parabolic, the absorption coefficient and optical band of materials can be determined using the Tauc relation [37,38,39]:(1)$$\alpha h\nu =A{\left(h\nu -E_{g}\right)}^{n}$$
where α is a linear absorption coefficient of a material, *h* is the Plank constant, *ν* is the light frequency, *A* is a proportionally coefficient characteristic of each material, and *n* is a constant associated with electron transition type [37,38,39]. For materials with a direct band gap *n* = ½ [31,32,33,34,37,38,39]. The values of the optical band gap of CaMnGdMoWO samples were obtained by extrapolating a linear part of (*αhν*)^2^ curve of each material to the photon energy axis as it is shown in Figure 7 b and Figure 8 b [24,29,30,31,32,33,36]. The determined *E_g_ values* are given in Table A1. Figure 9a,b shows the variation of *E_g_ values* with Gd^3+^ ions concentration (when *y* = 0.0200) of materials under study. It was observed that the optical band gap of CaMnGdMoWO samples decreased when Gd^3+^ concentration increased up to *x* = 0.2222 (solid state reaction method) or *x* = 0.1667 (combustion synthesis) and it reached the lowest values, i.e., 3.45 and 3.98 eV, respectively, whereby, the *E_g_ values* values determined for the materials with smaller grain size were clearly higher. We have already observed an analogous relationship for other nano- and micromaterials, i.e., Ca_1−*x*_Mn*_x_*MoO_4_ and Ca_1−*x*_Mn*_x_*(MoO_4_)_0.50_(WO_4_)_0.50_ solid solutions [30]. For example, the direct band gap estimated for Ca_0.95_Mn_0.05_MoO_4_ and Ca_0.95_Mn_0.05_(MoO_4_)_0.50_(WO_4_)_0.50_ microcrystals was 3.76 and 3.85 eV, respectively [30]. Analogous materials obtained by the combustion method showed a much higher *E_g_ values* value, i.e., Mn^2+^-doped calcium molybdate—4.18 eV and Mn^2+^-doped calcium molybdato-tungstate—4.07 eV [30].

The systematic decrease of the optical band with Gd^3+^-doping is attributed to structural disorders and one-site fluctuations, which arise due to the substitution of W^6+^ for Mo^6+^, Mn^2+^ and Gd^3+^ ions for Ca^2+^ ones. This last replacement is non-equivalent and goes according to the equation:3 Ca^2+^ → 2 Gd^3+^ + ▯(2)
where ▯ means vacant sites in cationic sublattice. Furthermore, there is a quite significant electronegativity difference between Ca (1.00), Gd (1.20), and Mn (1.55) as well as between Mo (2.16) and W (2.36). The differences in values electronegativity moved the valance band towards the conduction band and lead to a decrease in band gap with doping. For the CaMnGdMoWO materials with big Gd^3+^ ions concentration, i.e., when *x* > 0.2222 (samples obtained by high-temperature sintering) and *x* > 0.1667 (materials obtained by combustion method), direct band gap systematically increases. This means that defects in the CaMnGdMoWO crystal lattice are ordered. This structure is rearranged so as to be stable and regular. On the other hand, the band gap values determined for CaMnGdMoWO samples, in which the Mn^2+^ ions concentration changed, non-linearly decreased with increasing concentration of these ions (Figure 9c).

Hybridization properties of doping materials cause lattice strain and its defects, e.g., Mn *3d* electrons in CaMnGdMoWO can hybridize with O *2p* electrons and form intermediate states between valance and conduction bands. Band tails show a strong dependence on the localized states related to disorder in crystalline materials. The disorder is visible from the tailing of band edges and is estimated by Urbach energy (*E_U_*) [40,41,42,43,44,45,46]:(3)$$\alpha =\alpha_{0}\exp\left(\frac{h\nu }{E_{U}}\right)$$
where *α* is an experimentally determined absorption coefficient and *α*_0_ is a constant characteristic of the material. Due to the correlation of the band gap to the tailing of band edges, *E_g_* values reduce as *E_U_* values increase. Hence, these are complementary to each other. The Urbach energy is estimated by plotting ln(*α*) vs. *hν* and fitting the linear portion of the curve with a straight line [40,41,42,43,44,45,46]. The reciprocal of the slope of this linear region yields the *E_U_* value. The Urbach energy values determined for CaMoO_4_ and CaMnGdMoWO samples are collected in Table A1. Figure 9a–c shows *E_U_* variation as a function of Gd^3+^ (*x*) or Mn^2+^ (*y*) concentration. As is clearly seen from these figures, the optical band gap values change opposite to the degree of disorder in the structure of CaMnGdMoWO materials. As a result, both a decrease in the optical band gap and the broadening of the Urbach tail occurred. The lowest value of *E_g_* (3.45 eV) was observed when the Urbach energy reached its highest value, i.e., 541 meV for CaMnGdMoWO sample when *x* = 0.2222 and *y* = 0.0200.

### 2.4. EPR Spectra of CaMnGdMoWO Materials

Samples of CaMnGdMoWO solid solution were also investigated in the electron paramagnetic resonance (EPR) experiment. Considering the doping metallic ions and the used synthesis method, CaMnGdMoWO samples were divided into the following three groups:-obtained using solid-state reaction method (fixed Mn^2+^ contribution, i.e., *y* = 0.0200),-obtained using solid sta-te reaction method (fixed Gd^3+^ contribution, i.e., *x* = 0.1667) and,-obtained using combustion synthesis (fixed Mn^2+^ contribution, i.e., *y* = 0.0200).

EPR spectra of different samples were observed, including their evolution as a function of temperature within a temperature range of 80–300 K. Figure 10a shows resonance results for samples with fixed Mn^2+^ doping and varied gadolinium ions content, denoted by the *x* parameter. As can be seen, EPR spectra revealed the existence of multi-line agreement of manganese signal centered at *c.a.* 340 mT. This signal is clearly visible for samples with lower gadolinium ions content (*x =* 0.0050), where only six narrow lines of Mn^2+^ paramagnetic ions appear (the lowest spectrum in Figure 10a). Such a signal is characteristic for the condition of well-resolved hyperfine structure of manganese ions with the nuclear spin *I = 5/2*.

With increasing of Gd^3+^ doping, manganese signal starts to be disturbed, and finally, when gadolinium ions content exceeds *x* = 0.1430, only a wide gadolinium signal is observed near 340 mT magnetic field position (upper spectrum in Figure 10a).

Changes in the observed EPR signal is caused by the coexistence of magnetic Mn^2+^ with Gd^3+^ ions. This circumstance leads to perturbation of Mn^2+^ arrangement, where responsible magnetic centers no more fulfill the simple model of well-isolated ions. With increasing of gadolinium ions content, the overall EPR signal should be described rather as a combination of gadolinium and manganese magnetic arrangement with complex mutual interactions. At least, if gadolinium ions doping exceeds some level (in our case *x* > 0.1430) only a wide resonance line of Gd^3+^ ions is observed, as a result of the crystal field (fine) interaction of S = 7/2 electronic spin of gadolinium.

But influence of Gd^3+^ ions on the Mn^2+^ magnetic system originates not only from the paramagnetic nature of three-valent gadolinium ions. As we mentioned in our previous work, where comparable scheelites Ca_1−3_*_x_*_−*y*_Mn*_y_*▯*_x_*Eu_2_*_x_*(MoO_4_)_1−3_*_x_*(WO_4_)_3_*_x_* were studied [29], disturbance of Mn^2+^ is visible also under coexistence with non-magnetic Eu^3+^ ions. This mechanism was explained by charge non-equilibrium between di- and three-valent ions where the creation of additional electronic vacancies (▯*_x_*) takes place. An increasing number of electronic vacancies have an influence on nominally isolated manganese ions, leading to the creation of complex magnetic arrangements. Disturbance of the magnetic system is reflected in the evolution of the Mn^2+^ EPR spectra where, at a significantly high *x* value, only a wide overleaped resonance line is observed. Thus, as we believe, a similar mechanism is responsible for the evolution of manganese resonance signal in present materials. Overleaping of the manganese line leads to circumstances, where the gadolinium EPR signal dominates in the overall spectrum at a high *x* value.

As mentioned, CaMnGdMoWO samples were further investigated as a function of temperature. EPR signal decreases with increasing of temperature according to Curie-Weiss law: *I= C/(T-θ),* where *I* is the integral intensity of the resonance signal. Calculated values of the *θ* parameter are negative in all cases, indicating an antiferromagnetic (AFM) interaction among responsible magnetic centers: Mn^2+^ and Gd^3+^. The negative sign of magnetic interactions could be visible in a graph with the relation of *I·T* product as a function of temperature (Figure 10b). These diagrams show positive loop inclination between calculated values, for different manganese ions doping, which confirms the domination of AFM interactions among responsible ions.

Detailed calculations of *θ* parameter values are not presented, as being uncertain due to the mixing contribution of two different magnetic ions, but generally, *θ* parameter rather does not exceed a range of teens Kelvin’s, which indicates a not so strong (moderate) power of magnetic interactions in CaMnGdMoWO materials.

If gadolinium ions’ contribution exceed some level, the overall EPR signal consists of only a symmetric wide line ascribed to Gd^3+^ ions. Such a situation was clearly observed in the second group of GdMnGdMoWO materials, i.e., samples with fixed gadolinium ions content when *x* = 0.1667. The spectra shown in Figure 11a revealed the wide EPR line only, independentl of the variation of Mn^2+^-doping. Only for the sample with the smallest manganese concentration, i.e., when *y* = 0.0066*,* the trace of the Mn^2+^ signal is hardly visible in the spectrum (inset in Figure 11a).

Among presented three spectra the signal for *y* = 0.0667 seems to possess most symmetric shape. For this line detailed analysis of its shape has proceeded. Usually, symmetric EPR lines are successfully described by the Lorentzian function. In our case best results were obtained with using a small modification of the Lorentzian line known as the Dysonian function. According to this attitude, the resonance mechanism includes not only microwave absorption, but also some contribution to the dispersion process. As one can see, presented in Figure 11b simulated Dysonian line truly reflects the experimental line shape.

Figure 12a shows the calculated *I·T* product as a function of temperature for second group of CaMnGdMoWO samples with varied manganese ions doping. Positive loops, confirming domination of the AFM character of magnetic interaction among responsible magnetic ions, similarly as for group of samples with established manganese ions doping.

The last part of our investigations concerned the group of samples obtained by combustion method, with fixed Mn^2+^ content (*y =* 0.0200) and varied gadolinium ions contribution (*x*). EPR spectra of two samples with: *x =* 0.0455 ((3) line) and *x =* 0.2000 ((1) line) are presented in Figure 12b. As could be seen, with increasing Gd^3+^ content, the magnetic structure of present materials is modified, where the EPR signal of Mn^2+^ ions is significantly overleaped by a wide gadolinium line. It could be explained by the increasing role of exchange interactions between two magnetic centers: tetrahedral Mn^2+^ and dodecahedral Gd^3+^ ions taking place via an oxygen bridge. Thus, at higher Gd^3+^ concentrations, the manganese ions are affected by the environment, and could not be treated as isolated particles anymore.

Figure 12b additionally presents EPR spectra of samples synthesized by solid state reaction method. Comparing samples with similar *x* parameters one can conclude that EPR signals are almost unchanged, in respect to the synthesis method. As we reported in earlier work [30], synthesis during high-temperature sintering or combustion method modifies the magnetic properties of scheelites, which is expressed with changes in the EPR spectra. For Ca_1−*x*_Mn*_x_*MoO_4_ nanomaterials, combusted at 450 °C we observed relatively well-resolved manganese EPR lines. According to us, it was connected with a low size of grains in these materials, where the skin effect prevent the development of the magnetic interaction along the structure, allowing the maintenance Mn^2+^ ions as isolated centers with visible fine structure. On the other hand, EPR signal of combusted samples: Ca_1-*x*_Mn*_x_*(MoO_4_)_0.50_(WO_4_)_0.50_ was significantly overleaped, compared to a similar one obtained by solid–solid reaction. As these samples were synthesized at a combustion temperature 900 °C, we proposed an explanation with a mechanism, where a higher temperature process prefers the creation of manganese-enriched areas embedded in the matrix deprived of these ions. Thus, we observed that the EPR signal originates from a dense Mn^2+^ system, being far from an isolated ion model.

If we suppose, that the shape of EPR spectra is caused by competition between grain size and temperature of the thermal process, in the present case both mentioned factors lead to a specific equilibrium, where no changes in EPR spectra are observed (Figure 12b). It means, that relatively high combustion temperature (900 °C) in CaMnGdMoWO samples caused similar Mn^2+^ ions distribution in low grain materials, as in higher grains CaMnGdMoWO obtained by solid–solid reaction at a lower temperature. A similar weak influence of the thermal process on the EPR spectra wasobserved in our previous studies on Pb_1−3*x*_▯*_x_*Gd_2*x*_(MoO_4_)_1−3*x*_(WO_4_)_3*x*_ (0 < x ≤ 0.1774) scheelite structures [47].

## 3. Materials and Methods

### 3.1. Synthesis of CaMnGdMoWO Solid Solution

Microcrystalline samples of the new solid solution were obtained by two-step synthesis. In both steps, a high-temperature solid-state reaction between appropriate reactants was applied. The following initial materials were used in the first step of synthesis: calcium carbonate (CaCO_3_), manganese oxide (MnO), molybdenum oxide (MoO_3_), gadolinium oxide (Gd_2_O_3_), and tungsten oxide (WO_3_) (all raw materials of high-purity grade min. 99.95%, Alfa Aesar and without thermal pre-treatment). Calcium molybdate (CaMoO_4_), manganese molybdate (MnMoO_4_), and gadolinium tungstate (Gd_2_(WO_4_)_3_) were obtained analogously to our previous studies [26,29]. In the next step, two series of ternary mixtures comprising CaMoO_4_, MnMoO_4_, and Gd_2_(WO_4_)_3_ were prepared, i.e., when MnMoO_4_ concentration was constant and equaled 3.00 mol% (*y* = 0.0200) and gadolinium tungstate was variable and ranged from 0.50 (*x* = 0.0050) to 50.00 mol% (*x* = 0.2500). The initial MnMoO_4_/Gd_2_(WO_4_)_3_/CaMoO_4_ mixtures of the second series contained constant amount of Gd_2_(WO_4_)_3_, i.e., 25.00 mol% (*x* = 0.1667), and the content of MnMoO_4_ varied from 1.00 (*y* = 0.0066) to 10.00 mol% (*y* = 0.0667). The composition of all prepared mixtures is given in Table A1 (Appendix A). They were sintered in corundum crucibles, in several 12 h heating stages, in air atmosphere, and at temperatures ranging from 900 to max. 1160 °C. After each sintering period, the mixtures were slowly cooled down to ambient temperature, weighed, homogenized in a porcelain mortar, and examined using the X-ray diffraction (XRD) method. A slight mass loss (not higher than 0.09%) was observed for each initial mixture during its heating. This observation clearly shows that the synthesis of new CaMnGdMoWO solid solution runs practically without mass change. This process can be described according to the general equation:(1 − 3*x* − *y*) CaMoO_4_ + *x* Gd_2_(WO_4_)_3_ + *y* MnMoO_4_ = Ca_1−3*x*−*y*_Mn*_y_*▯*_x_*Gd_2*x*_(MoO_4_)_1−3*x*_(WO_4_)_3*x*_(4)

The formula of each obtained sample is shown in Table A1.

Some samples of CaMnGdMoWO solid solution were synthesized via the combustion method [27]. As precursors in this synthesis, the following high-purity reactants were used: Gd_2_O_3_, MnO, CaCO_3_, ammonium molybdate ((NH_4_)_6_Mo_7_O_24_·1.36 H_2_O, Alfa Aeasar), ammonium tungstate (NH_4_)_10_W_12_H_2_O_42_·3.45 H_2_O, Alfa Aeasar) and citric acid (as fuel, C_6_H_8_O_7_·H_2_O, Alfa Aeasar) [27]. The Gd, Mn, and Ca precursors as well as fuel were dissolved on heating in an aqueous solution of nitric acid (1:1). Next, ammonia solution was slowly added, and pH was made to ~5. Appropriate amounts of Mo and W precursors were dissolved in hot water in a separate beaker. The obtained solutions were mixed together and heated at 110 °C to completely evaporate water. In the next step, the obtained gel was heated carefully at 300 °C. During combustion synthesis the gel burned out with rapid evolution of a large quantity of fume, yielding voluminous powder. Finally, the as-burnt powder was heated at 900 °C for 2 h [27].

### 3.2. Characterization of Methods

Powder X-ray diffraction patterns of all samples were recorded within the 10–100° *2Θ* range with the scanning step 0.013° on an EMPYREAN II diffractometer (PANalytical, Almelo, The Netherlands) using CuKα_1,2_ radiation (λ = 1.5418 Å). Next, XRD patterns were analyzed by HighScore Plus 4.0 software, and lattice parameters were calculated using POWDER 2.0 software [48,49]. Density of each sample was measured on a Quantachrome Instruments Ultrapycnometer (model Ultrapyc 1200 e, Boynton Beach, FL, USA) using nitrogen (purity 99.99%) as a picnometric gas.

Simultaneous TG and DTA measurements (the results not presented here) of ammonium molybdate and tungstate were carried out on a TA Instruments thermoanalyzer (model SDT 2960, USA) at the heating rate of 10 deg min^−1^, and in the temperature range from 25 to 700 °C (the air flow 110 mL h^−1^). The mass of each sample for DTA-TG measurements was ~30 mg. The mass losses recorded on TG curves and connected with the dehydration processes of both hydrates allowed a precise determination of the number of water molecules in Mo and W precursors.

Melting point of some CaMnGdMoWO samples was determined using a pyrometric method. The samples of the solid solution when *x* = 0.0050; 0.1667; 0.2500 and *y* = 0.0200 as well as *x* = 0.1667 and *y* = 0.0667, previously pressed into pellets using a hydraulic press at 15 MPa, were heated in a resistance furnace. Their image and temperature were continuously recorded by a pyrometer Raytek (model RAYMM1MHSF2V) during the gradual rise of temperature in a furnace. The melting point of each sample was determined at this temperature when its image of pellets disappeared in a camera. This moment indicated that the pellet had melted.

UV-vis diffuse reflectance spectra were recorded within the range of 200–1000 nm using a JASCO-V670 spectrophotometer (JASCO Europe S.R.L., Cremella, Italy) equipped with an integrating sphere.

Grain size and morphology of some CaMnGdMoWO materials as well as their elemental composition were examined using field emission scanning electron microscopy (FE-SEM) Hitachi SU–70 (Hitachi, Naka, Japan) microscope and NORAN™ System 7 of Thermo Fisher Scientific (Madison, WI, USA) equipped with UltraDry energy dispersive X-ray detector), respectively. SEM analysis was performed at an accelerating voltage of 10kV and secondary electron images were acquired. The samples under study were coated with a palladium–gold alloy thin film using the thermal evaporation PVD method (evaporator JEOL JEE-4X (JEOL, Tokyo, Japan) to provide electric conductivity.

EPR spectra of doped materials were recorded on a conventional X-band Bruker ELEXSYS E 500 CW-spectrometer operating at 9.5 GHz with 100 kHz magnetic field modulation. The first derivative of the absorption spectra has been recorded as a function of the applied magnetic field. Temperature dependence of the EPR spectra of solid solutions under study in the 78–300 K temperature range was recorded using Oxford Instruments ESP helium-flow cryostat.

## 4. Conclusions

In this study, new Ca_1−3*x*−*y*_Mn*_y_*▯*_x_*Gd_2*x*_(MoO_4_)_1−3*x*_(WO_4_)_3*x*_ solid solution (*x* = 0.0050; 0.0098; 0.0283; 0.0455; 0.0839; 0.1430; 0.1667; 0.2000; 0.2222; 0.2500 when *y* = 0.0200 as well as *y* = 0.0066; 0.0333; 0.0667 when *x* = 0.1667) was synthesized using solid state reaction method and combustion route. XRD analysis identified that new ceramic materials tend to crystallize in tetragonal scheelite-type structure (space group *I*4_1_/*a*) within the whole homogeneity of solid solution. Unexpectedly, both lattice parameters (*a* and *c*) changed nonlinearly (usually increased) with increasing Gd^3+^-doping when Mn^2+^ ions content in the samples was constant. Expansion of the unit cell volume was observed practically within a whole homogeneity concentration range of solid solution. New ceramics melt in air and their melting point decreased as Gd^3+^ as well as Mn^2+^ ions concentrations increased. Applying of combustion method made it possible to obtain CaMnGdMoWO powders with a submicro grain size, i.e., from ~500 nm to max. a few micrometers. All materials are insulators and their direct band gap nonlinearly changed with the increase of both doping ions’ concentrations. The UV-vis studies revealed a structural disorder in CaMnGdMoWO samples that was confirmed by an increase of Urbach energy with Gd^3+^ and Mn^2+^-substitution.

EPR spectra of CaMnGdMoWO materials confirmed the existence of two types of magnetic objects, i.e., Mn^2+^ and Gd^3+^ ions. The fine structure of six well-resolved manganese ions was observed only in samples with very low gadolinium ions contents. Under these circumstances manganese ions occupy a central position in MoO_4_ tetrahedra. Increasing concentration of Gd^3+^ ions in the dodecahedral position disturbs the Mn^2+^ arrangement, which means that manganese should not be treated as well-isolated ions, and MoO_4_ tetrahedra are significantly distorted. The fine structure of Mn^2+^ starts to be overleaped by complex dipolar and exchange interactions thus, as a result, only a wide gadolinium ions resonance line is visible in EPR spectra at higher Gd^3+^ contents.

The temperature dependence of the EPR signal intensity fulfills the Curie–Weiss relation, indicating significant antiferromagnetic (AFM) interactions among responsible ions. Detailed analysis is difficult due to the combined coexistence of two different magnetic Mn^2+^and Gd^3+^ ions in this case. But the general domination of AFM interactions is proved and similar results were reported earlier, where Mn^2+^ ions were embedded in comparable scheelite materials.

A significantly lower grain size of samples obtained by combustion route is the reason of maintaining Mn^2+^ ions as being better isolated in structure, compared to samples obtained by solid–solid reactions. On the other hand, a high temperature of combustion could be the reason for the non-uniform distribution of Mn^2+^ ions, where mutual interactions are conducive to creating complex magnetic arrangements.

## Figures and Tables

**Figure 1 ijms-23-15740-f001:**
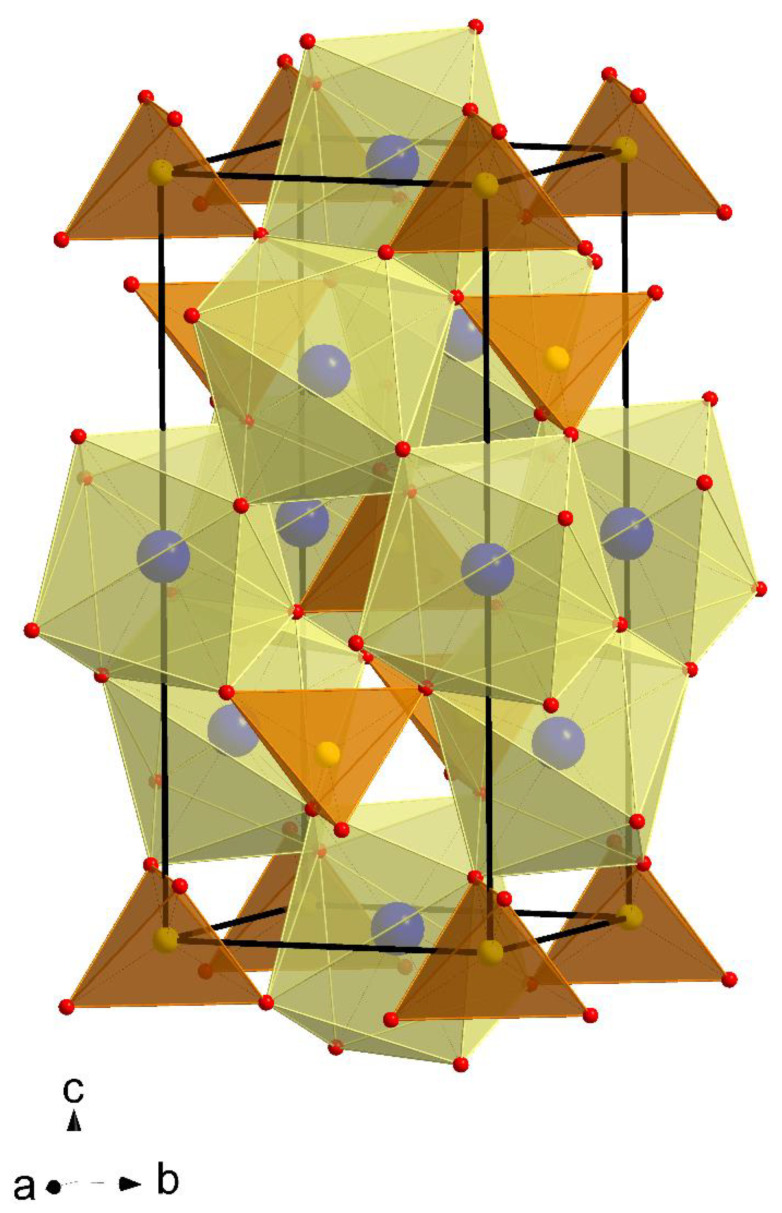
Structure of CaMoO_4_ (Ca^2+^—blue circles; Mo^6+^—yellow circles; O^2–^—red circles).

**Figure 2 ijms-23-15740-f002:**
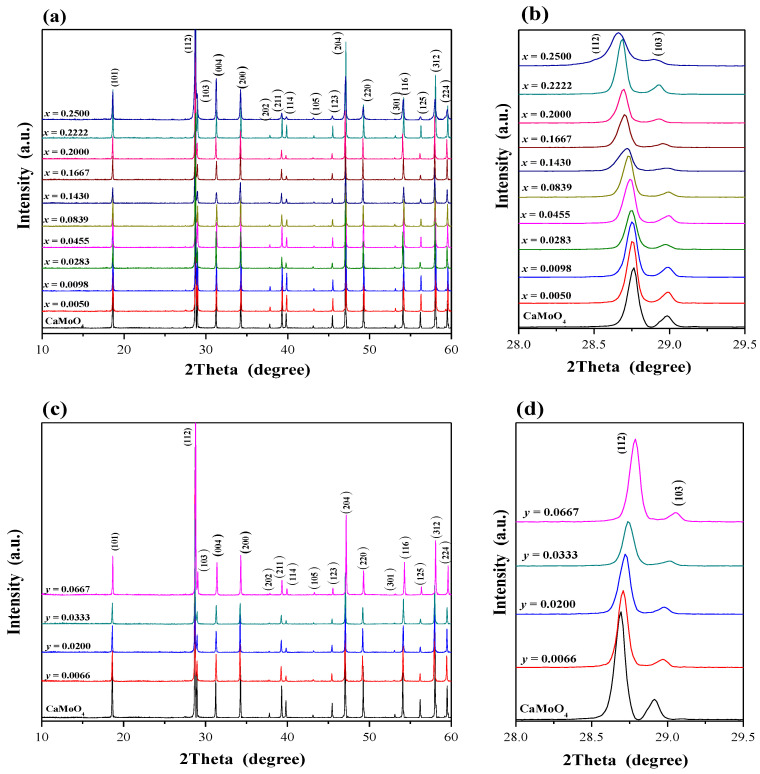
XRD patterns of CaMoO_4_ and CaMnGdMoWO samples when 0 < *x* ≤ 0.2500 and *y* = 0.0020 (**a**,**b**)—solid-state reaction method; (**e**,**f**)—combustion route; when 0 < *y* ≤ 0.0667 and *x* = 0.1667 (**c**,**d**)—solid- state reaction method.

**Figure 3 ijms-23-15740-f003:**
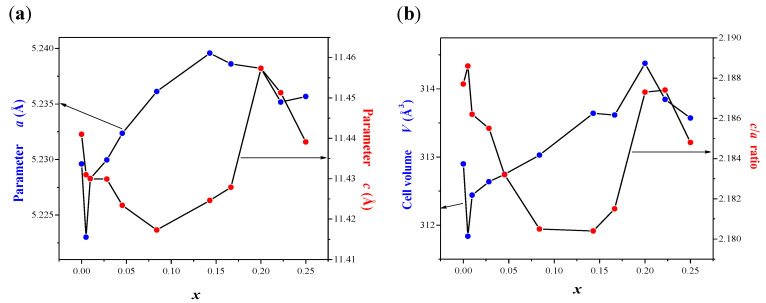
Variation of *a* and *c* unit cell parameters (**a**) as well as volume of unit cell (*V*) and lattice parameter ratio *c*/*a* (**b**) of CaMnGdMoWO samples obtained by high-temperature sintering as a function of *x* value.

**Figure 4 ijms-23-15740-f004:**
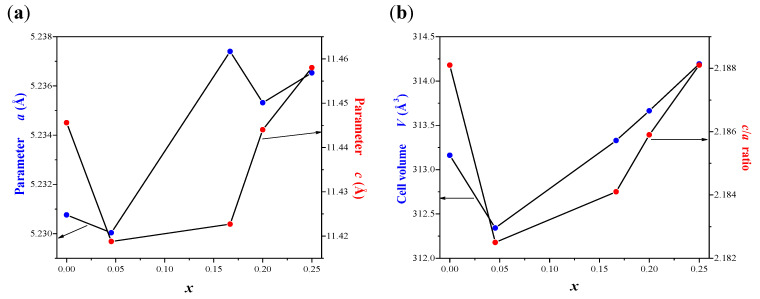
Variation of *a* and *c* unit cell parameters (**a**) as well as the volume of unit cell (*V*) and lattice parameter ratio *c*/*a* (**b**) of CaMnGdMoWO samples obtained by combustion route as a function of *x* value.

**Figure 5 ijms-23-15740-f005:**
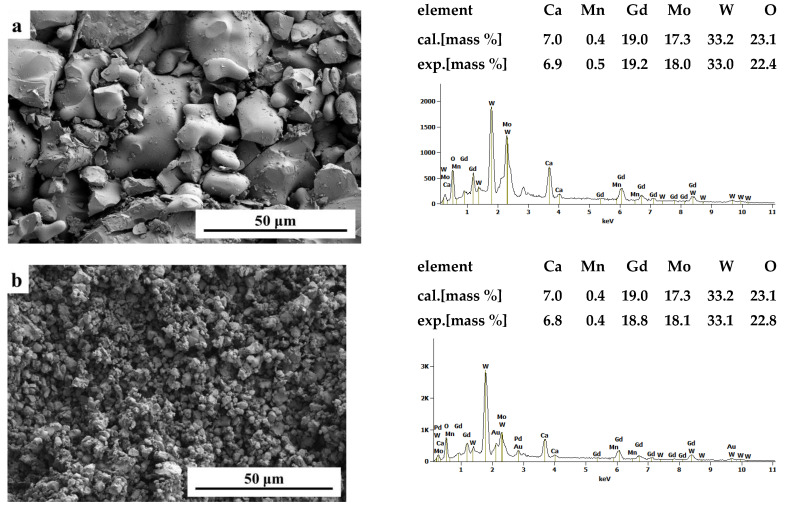
SEM (secondary electrons) images and EDS elemental analysis of CaMnGdMoWO samples when *x* = 0.1667 and *y* = 0.0200 obtained by high-temperature sintering (**a**) and combustion method (**b**).

**Figure 6 ijms-23-15740-f006:**
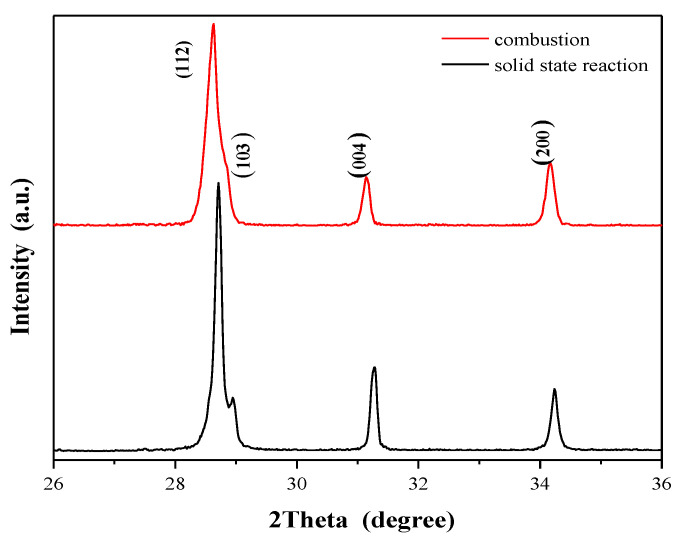
XRD patterns of CaMnGdMoWO samples when *x* = 0.2500 and *y* = 0.0020 obtained by solid-state reaction method (black line) and combustion route (red line) within the range of *2Θ* from 26.0 to 36.0°.

**Figure 7 ijms-23-15740-f007:**
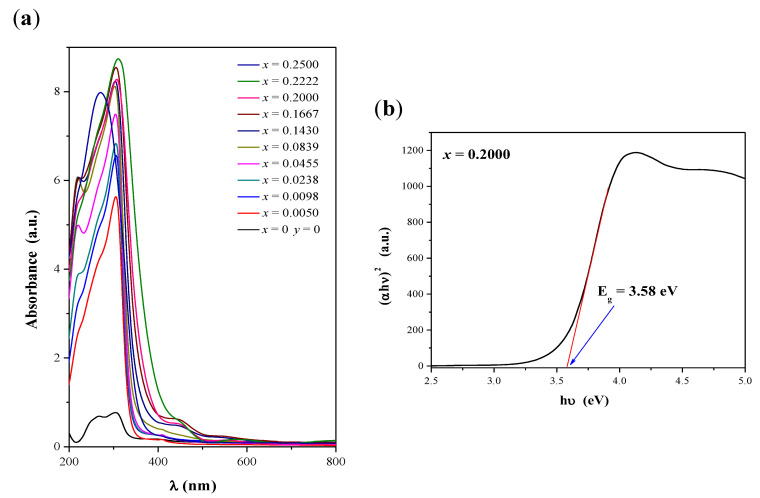
UV-vis absorption spectra of CaMoO_4_ and CaMnGdMoWO ceramic materials obtained by solid state reaction method when 0 < *x* ≤ 0.2500 and *y* = 0.0200 (**a**); Plot of (αhν)^2^ vs. photon energy (hν) of the solid solution sample when *x* = 0.2000 and *y* = 0.0200 with determined band gap energy (*E_g_*) (**b**).

**Figure 8 ijms-23-15740-f008:**
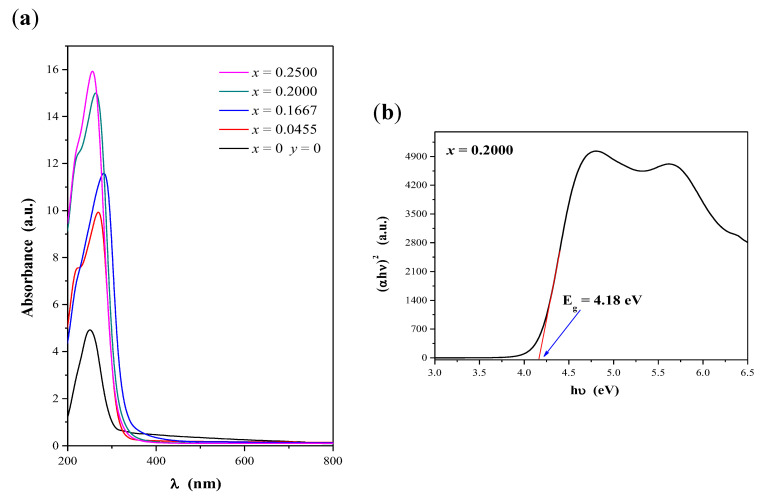
UV-vis absorption spectra of CaMoO_4_ and CaMnGdMoWO ceramic materials obtained by combustion method when 0 < *x* ≤ 0.2500 and *y* = 0.0200 (**a**); Plot of (αhν)^2^ vs. photon energy (hν) of the solid solution sample when *x* = 0.2000 and *y* = 0.0200 with determined band gap energy (E_g_) (**b**).

**Figure 9 ijms-23-15740-f009:**
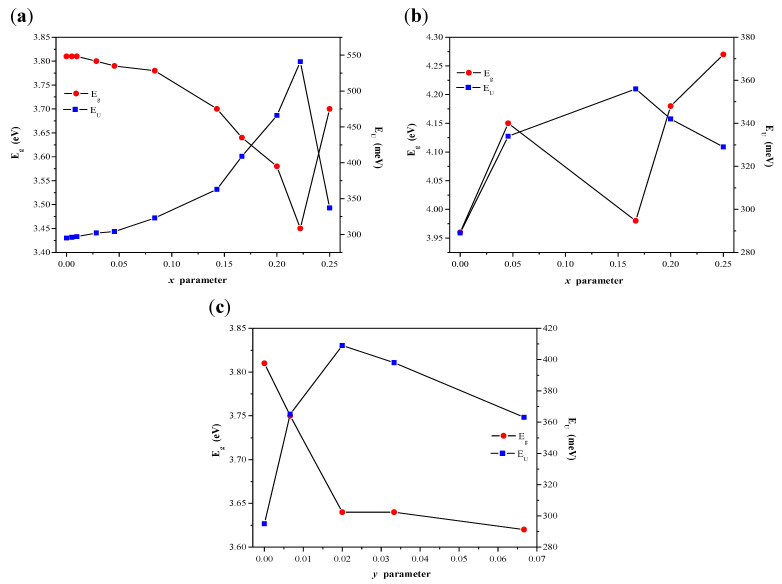
Variation of direct band gap (*E_g_*) and Urbach energy (*E_U_*) with Gd^3+^ (*x*) or Mn^2+^ (*y*) doping in CaMnGdMoWO ceramic materials obtained by solid-state reaction method (**a**,**c**) and combustion route (**b**).

**Figure 10 ijms-23-15740-f010:**
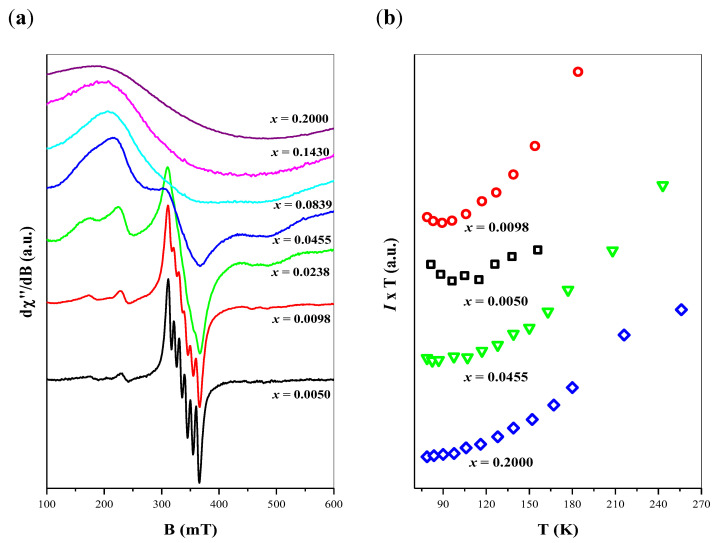
EPR spectra of CaMnGdMoWO samples with varied Gd^3+^ doping obtained by solid state reaction method (**a**); Temperature variation of *I^.^T* product, where *I*- integral intensity of EPR signal (**b**).

**Figure 11 ijms-23-15740-f011:**
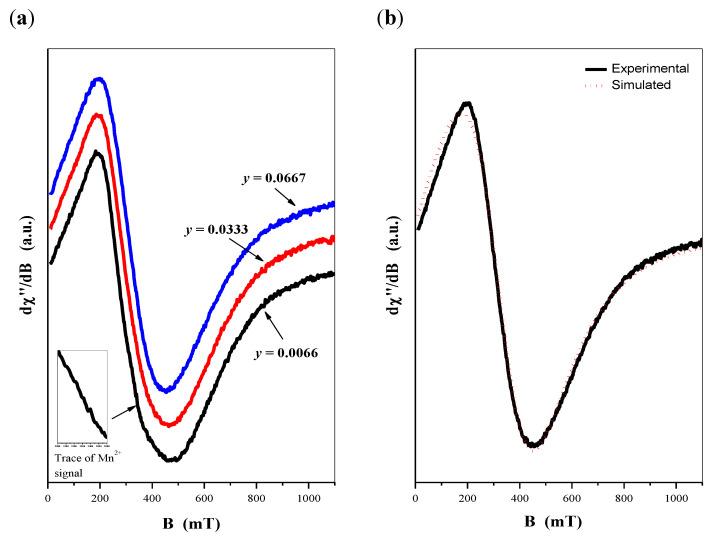
EPR spectra of CaMnGdMoWO samples with fixed Gd^3+^ content *x* = 0.1667 obtained by solid state reaction method (**a**); EPR spectrum of CaMnGdMoWO sample with *y* = 0.0667 and its simulation by Dysonian function (**b**).

**Figure 12 ijms-23-15740-f012:**
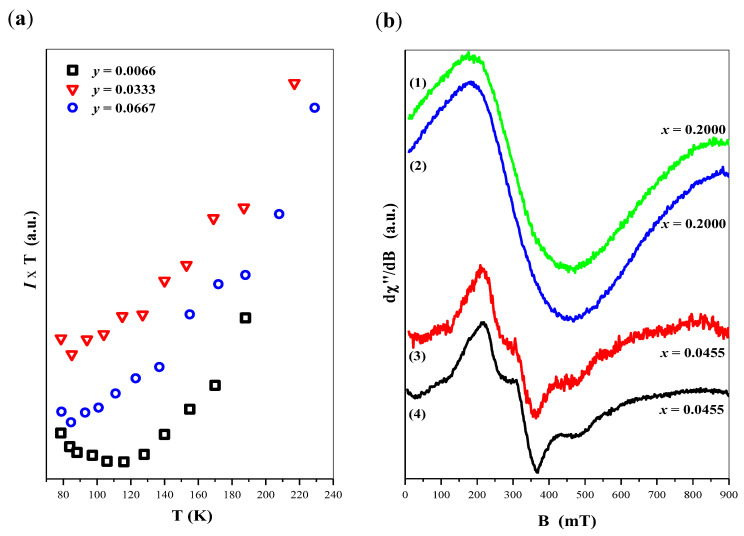
Temperature variation of *I^.^T* product of CaMnGdMoWO materials when *x* = 0.1667 (**a**); room temperature EPR signal of CaMnGdMoWO samples when *y* = 0.0200 obtained by solid state reaction ((1) and (3) line) and combustion ((2) and (4) line) methods (**b**).

## Data Availability

Not applicable.

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
