# Peer review of "New Gd3+ and Mn2+-Co-Doped Scheelite-Type Ceramics—Their Structural, Optical and Magnetic Properties"

_ijms, 2022, doi:10.3390/ijms232415740_

Round 1

Reviewer 1 Report

Gd3+ and Mn2+ co-doped CaMo(W)O4 phosphors were prepared by solid state method and combustion method in the work. The effects of Gd3+ content, Mn2+ content and synthesis method on the lattice parameters, powder size, band gap (Eg), Urbach energy (EU) and magnetic properties of the phosphors were studied in detail. The samples synthesized by combustion method have more uniform and fine particle size, and there is obvious antiferromagnetic interaction between Gd3+ and Mn2+. Interestingly, no significant change in the EPR spectra of the two groups of phosphors may be attributed to the special distribution mechanism of Mn2+. To confirm this conclusion requires the authors to explore more deeply. The author's language and data display are worthy of recognition, but the authors still need to correct and clarify some issues. In addition, there are also small errors in the content and format of the text. The manuscript can be recommended for publication after a major revision. These questions are as follows: 

1. The authors said that the synthesized phosphor has a scheelite-type structure. Is there any relevant proof? Please explain. 

2. The authors said that with the increase of Gd3+ and Mn2+ content, the XRD diffraction patterns shift to higher and lower angles, respectively, but it is difficult to see in Fig.2. Therefore, it is more appropriate to add the enlarged diagram of the dominant peak. 

3. The authors said that compared to the a parameter, the c parameter shows an opposite Gd3+ concentration dependence. But actually, the turning points of a and c parameters are not consistent, and their trends are complex. The author's statement is not appropriate. 

4. The authors had mentioned Table S1 many times, and there seems to be a lot of crystal structure information in Table S1, but there is no Table S1 in the manuscript uploaded by the authors. The authors need to supplement it and make sure there are no errors. 

5. The EDX spectra was mentioned by the authors, but not in the manuscript. Please supplement it. 

6. The authors believed that the Eg values determined for the material with smaller grain size were clearly higher. Whether this conclusion is confirmed in other studies, please quote literatures to demonstrate it. In addition, do the authors consider the effect of crystal structure differences on Eg in this work? 

7. In the caption of Fig. 12, do the authors mean “black and blue line”? Please correct it. 

8. The authors believed that the reason for the unchanged EPR spectra of the samples synthesized by solid state method and combustion method in this work may be due to the similar distribution mechanism of manganese ions. Do you mean that the distribution mechanism of manganese ions is the essential reason for the difference in EPR spectra, not the difference in particle size caused by different synthesis methods? Is this credible? Is it inconsistent with the previous work conclusions? 

9. What is the relationship between crystal structure and magnetic objects?

Author Response

Response to Reviewer 1 Comments

Point 1: The authors said that the synthesized phosphor has a scheelite-type structure. Is there any relevant proof ? Please explain.

Response 1:  As it is shown in Fig. 2, on the powder diffraction patterns of materials under study, only peaks from scheelite type structure were registered. All diffraction lines were assigned Miller indices from CaMoO4 set (JCPDs No. 04-013-6763). Indexing procedures were carried out only for the tetragonal system and the obtained solutions corresponded to space group I41/a.

Point 2: The authors said that with the increase of Gd3+ and Mn2+ content, the XRD diffraction patterns shift to higher and lower angles, respectively, but it is difficult to see in Fig.2. Therefore, it is more appropriate to add the enlarged diagram of the dominant peak. 

Response 2: As suggested by the reviewer, we modified Figure 2, i.e. the enlarged portion of XRD patterns from 28 to 29.5° 2Q with (112) and (103) diffraction peaks are presented in Figures 2 b, d and f.

Point 3: The authors said that compared to the a parameter, the c parameter shows an opposite Gd3+ concentration dependence. But actually, the turning points of a and c parameters are not consistent, and their trends are complex. The author's statement is not appropriate. 

Response 3:  We have modified the text on changes of a and c lattice parameters as follows: Initially, when Gd3+ content is relatively low, the a parameter increases up to 5.23958(6) Å (x = 0.1430, synthesis by solid state reaction) or 5.2374740(8) Å (x = 0.1667, combustion) and then, its value decreases and then increases again with further increasing content of gadolinium ions. The c parameter shows a different relationship, i.e. for low Gd3+ ions concentrations its value decreases to 11.4173(7) Å (= 0.0839, solid state reaction method) or 11.4188(9) Å (x = 0.0455, combustion). With a further increase of Gd3+-doping, the value of c parameter increased and then decreased (high-temperature sintering) or only increased (combustion route).

Point 4: The authors had mentioned Table S1 many times, and there seems to be a lot of crystal structure information in Table S1, but there is no Table S1 in the manuscript uploaded by the authors. The authors need to supplement it and make sure there are no errors. 

Response 4:  Table S1 is included at the end of the manuscript as Appendix.

Point 5: The EDX spectra was mentioned by the authors, but not in the manuscript. Please supplement it. 

Response 5:  As suggested by the reviewer, the EDS spectra of both samples are shown in Figure 5. This Figure was significantly modified by us.

Point 6. The authors believed that the Eg values determined for the material with smaller grain size were clearly higher. Whether this conclusion is confirmed in other studies, please quote literatures to demonstrate it. In addition, do the authors consider the effect of crystal structure differences on Eg in this work? 

Response 6: We already observed higher energy gap values for nanomaterials in our earlier studies [M. Pawlikowska, H. Fuks, E. Tomaszewicz. Solid state and combustion synthesis of Mn2+-doped scheelites – Their optical and magnetic properties. Ceram. Int. 43 (20017) 14135-14145]. Optical band gap estimated for Ca1-xMnxMoO4 and Ca1-xMnx(MoO4)0.50(WO4)0.50 (x = 0.05) nanomaterials were 4.18 and 4.07 eV, respectively. The Eg values for analogous materials obtained by solid state reaction method were 3.76 eV (Mn2+-doped molybdates and x = 0.05) and 3.85 eV (Mn2+-doped molybdato-tungstates and x = 0.05). We mentioned these materials in the revised version of our manuscript.

Point 7:  In the caption of Fig. 12, do the authors mean “black and blue line”? Please correct it. 

Response 7: The caption of Figure 12 has been corrected by us. We have removed the notation: black and blue lines. The EPR spectra were marked with Arabic numbers.

Point 8: The authors believed that the reason for the unchanged EPR spectra of the samples synthesized by solid state method and combustion method in this work may be due to the similar distribution mechanism of manganese ions. Do you mean that the distribution mechanism of manganese ions is the essential reason for the difference in EPR spectra, not the difference in particle size caused by different synthesis methods? Is this credible? Is it inconsistent with the previous work conclusions ? 

Response 8: We answered the reviewer's questions by significantly modifying the text on EPR results.

Point 9: What is the relationship between crystal structure and magnetic objects?

Response 9: On the basis of EPR studies, we can infer only the closest surroundings of magnetic ions and their local symmetry.

Reviewer 2 Report

The synthesys and characterization of the scheelite-type ceramics are presented. Both solid state reaction and combustion methods are use to prepare the samples. XRD patterns of both are similar while SEM images shows differences in the size of grains. Optical studies reveal the presence of vacancies. EPR study supplies the infromation on the magnetic nature of Gd and Mn ions.

Therefore, the manuscript reports results of promissing new materials synthesis and their detailed characterization. It is sutable for publication in International Journal of Molecular Sciences. I advise the authors to revise English in the chapter 3.4 where in some places 'on' shoul be replaced with 'of', and replace square denoting the vacancy in the chemical formula with some Roman letter like X or D to improve readibility.

Author Response

Response to Reviewer 2 Comments

Point 1: The synthesys and characterization of the scheelite-type ceramics are presented. Both solid state reaction and combustion methods are use to prepare the samples. XRD patterns of both are similar while SEM images shows differences in the size of grains. Optical studies reveal the presence of vacancies. EPR study supplies the infromation on the magnetic nature of Gd and Mn ions.

Therefore, the manuscript reports results of promissing new materials synthesis and their detailed characterization. It is sutable for publication in International Journal of Molecular Sciences. I advise the authors to revise English in the chapter 3.4 where in some places 'on' shoul be replaced with 'of', and replace square denoting the vacancy in the chemical formula with some Roman letter like X or D to improve readibility.

Response 1: We have made some linguistic corrections. We think that use of a capital X for vacancies could be risky because we already applied a small x as a concentration parameter. On the other hand, a capital letter D may suggest a deuterium - one of hydrogen isotope. The current manuscript is 36th our work were we used this symbol for vacancied solution. It seems to us that for all readers who follow our studies to use this symbol will be more convenient. Therefore, we kindly ask the reviewer to leave the vacancy symbol ().
